

# Migration in the social stage of *Dictyostelium discoideum* amoebae impacts competition

Chandra N. Jack[1], Neil Buttery[2], Boahemaa Adu-Oppong[2], Michael Powers[3], Christopher R.L. Thompson[4], David C. Queller[2] and Joan E. Strassmann[2]

[1] Department of Plant Biology, Michigan State University, East Lansing, MI, United States of America
[2] Department of Biology, Washington University, St. Louis, United States of America
[3] Department of Biosciences, Rice University, Houston, United States of America
[4] Faculty of Life Sciences, The University of Manchester, Manchester, United Kingdom

Corresponding author
Chandra N. Jack,
chandra.jack@gmail.com

## ABSTRACT

Interaction conditions can change the balance of cooperation and conflict in multicellular groups. After aggregating together, cells of the social amoeba *Dictyostelium discoideum* may migrate as a group (known as a slug) to a new location. We consider this migration stage as an arena for social competition and conflict because the cells in the slug may not be from a genetically homogeneous population. In this study, we examined the interplay of two seemingly diametric actions, the solitary action of kin recognition and the collective action of slug migration in *D. discoideum*, to more fully understand the effects of social competition on fitness over the entire lifecycle. We compare slugs composed of either genetically homogenous or heterogeneous cells that have migrated or remained stationary in the social stage of the social amoeba *Dictyostelium discoideum*. After migration of chimeric slugs, we found that facultative cheating is reduced, where facultative cheating is defined as greater contribution to spore relative to stalk than found for that clone in the clonal state. In addition our results support previous findings that competitive interactions in chimeras diminish slug migration distance. Furthermore, fruiting bodies have shorter stalks after migration, even accounting for cell numbers at that time. Taken together, these results show that migration can alleviate the conflict of interests in heterogeneous slugs. It aligns their interest in finding a more advantageous place for dispersal, where shorter stalks suffice, which leads to a decrease in cheating behavior.

## INTRODUCTION

Individuals often interact with others in their environment, whether it is multicellular organisms such as lions in a plain, or bacteria in a wound in a more microscopic level. These interactions are characterized by both the effect on the recipient of an action and the effect of the behavior on the initiator of the action. For many years altruistic interactions,

those that benefit the recipient but impose a cost on the actor, confounded evolutionary biologists because they seem to provide a perfect setting for cheating, where individuals could gain the benefits of cooperative individuals without contributing to the public good (*Hamilton, 1964a*; *Hamilton, 1964b*; *Axelrod & Hamilton, 1981*; *Lehmann & Keller, 2006*; *Ghoul, Griffin & West, 2014*). In this setting, a cooperative population would be overcome with cheaters, leading to its collapse. However, *Hamilton (1964a)* showed that altruism could evolve if individuals preferentially directed benefits to kin. This theory, known as kin selection, requires individuals to be sufficiently related to overcome the costs of their cooperative behaviors.

At its face, kin selection, while a social behavior, is a solitary interaction between two individuals. Individual A senses individual B and based on some cue, be it genetic or environmental, either directs resources towards B or not (*Hurst & Beynon, 2010*; *Coffin, Watters & Mateo, 2011*; *Leclaire et al., 2013*). Yet there are many social behaviors that require the collective action of a group of individuals. In higher organisms, the flight patterns of migratory birds, group babysitting in meerkats, and schooling in fish are all examples of collective action. Just as in other social behaviors, they are mirrored in microbes. For example, there is swarming in *Myxoccocus xanthus* and fruiting body formation in *Dictyostelium* (*Crespi, 2001*; *Velicer & Yu, 2003*). In microbes, many studies have shown that relatedness is necessary for collective actions (*Ross-Gillespie & Kümmerli, 2014*). By studying both kin selection and collective behaviors in both higher organisms and microbes, we can gain a deeper understanding of the evolution of multicellularity, a collective action where independent individuals give up their own autonomy to form a higher-level group (*Szathmáry & Smith, 1995*; *Queller, 2000*).

*Dictyostelium discoideum* can be used for the study of both individual (kin recognition) and collective actions (development) making it ideal for the study of multicellularity. *D. discoideum* reproduces by binary fission and preys on soil bacteria. When resources become scarce, individuals send out a chemical signal that causes all nearby cells to aggregate together and initiate development. Once aggregated, the cells begin differentiating. The majority of the cells, approximately 80% will form reproductive spores while the remaining cells will altruistically form sterile stalk (*Kessin, 2001*). Unlike metazoans that go through a single-cell bottleneck at the zygote stage, *Dictyostelium* forms a metazoan-like aggregate that may be made up of several genotypes, thus providing an arena for competition, conflict, and manipulation.

Indeed, cheaters have been identified that are consistently over-represented in the sorus when mixed with another strain in both nature and in the laboratory setting (*Strassmann, Zhu & Queller, 2000*; *Fortunato, Queller & Strassmann, 2003*; *Queller et al., 2003*; *Gilbert et al., 2007*). However, all of these experiments were done bypassing a part of the lifecycle that involves another collective action—migration. If the present environment is not conducive to reproductive success, the group of cells, now known as a slug, can collectively migrate to a better location to finish development (*Kessin, 2001*). While it seems like there should not be any conflict within the slug, because this process allows cells to escape a poor environment, there is evidence of some conflict. *Foster et al. (2002)* found that clonal slugs

travel further than chimeric slugs composed of the same number of cells. This conflict could be avoided if the cells segregated to form separate slugs but experiments show that larger slugs move faster than smaller slugs and that a larger chimeric slug will travel further than smaller clonal ones, which could make a huge difference if a slug is attempting to reach a favorable location (*Inouye & Takeuchi, 1980*; *Foster et al., 2002*).

We know that there is competition between genotypes when there is no slug migration but why during the migration stage? The cells risk death if they aggregate in a location that is not conducive to dispersal and reproduction so why isn't there some type of armistice while migrating? It turns out that slug migration is costly (*Jack et al., 2011*). As the slug moves, prestalk cells are left behind in a slime trail (*Bonner, Koontz & Paton , 1953*; *Sternfeld, 1992*; *Kuzdzal-Fick et al., 2007*). The remaining cells must redifferentiate to maintain the proper slug proportioning of prestalk and prespore cells, which leads to a decrease in the number of reproductive spores that are formed (*Abe et al., 1994*; *Ràfols et al., 2001*; *Jack et al., 2011*). Decreasing the number of reproductive spores may set the stage for increased conflict if the slug is not homogeneous, similar to how limited resources may cause escalation of fights between higher organisms. For this reason, we predict that prolonging the time heterogeneous slugs migrate will accentuate competition because it prolongs the time genotypes compete against each other and decreases the availability of reproductive spores.

## MATERIALS & METHODS

### Growth and maintenance of strains

We used five naturally occurring clones of *D. discoideum* (*NC28.1*, *NC34.1*, *NC63.2*, *NC85.2* and *NC105.1*) originally collected in North Carolina (*Francis & Eisenberg, 1993*), which have been used in several previous studies on the social behavior of *D. discoideum* (e.g., *Fortunato, Queller & Strassmann, 2003*; *Buttery et al., 2009*). We grew spores from frozen stocks on SM agar plates (10 g peptone, 1 g yeast extract, 10 g glucose, 1.9 g $KH_2PO_4$, 1.3 g $K_2HPO_4$, 0.49 g $MgSO_4$ (anhydrous) and 17 g of agar per liter) in the presence of *Klebsiella aerogenes* (Ka) bacteria at a temperature of 22 °C.

### Transformation of wild clones

We collected actively growing and dividing cells from the edges of plaques grown in association with *Ka* on SM agar plates and transferred them to HL5 axenic medium (5 g proteose peptone, 5 g thiotone E peptone, 10 g glucose, 5 g yeast extract, 0.35 g $Na_2HPO_4 \cdot 7H_2O$, 0.35 g $KH_2PO_4$ per liter (*Watts & Ashworth, 1970*)) + 1% PVS (100,000 units of penicillin, 100 mg streptomycin sulphate, 200 μg folate, 600 μg vitamin B12 per liter) that was changed daily. The HL5 was changed daily until the culture dishes were free of visible bacteria. We then harvested the cells and washed them twice by centrifugation and resuspended them in cold standard KK2 buffer (16.1 mM $KH_2PO_4$ and 3.7 mM $K_2HPO_4$). Once the culture dishes were free of visible bacteria, we followed the procedure for the transformation of *D. discoideum* by (*Pang, Lynes & Knecht, 1999*) with red fluorescent protein (RFP) on an actin-15 promoter and a G418-resistance cassette.
The cells were transferred to culture dishes containing HL5 + 1% PVS and left overnight. After 24 h the medium was replaced with fresh medium containing 20 µg/ml G418 and changed daily for five days of selection. Wild *D. discoideum* clones do not grow well in axenic medium so we transferred the amoebae to SM agar with *Ka* to propagate. Plaques that fluoresced red under a (535 nm) light source were transferred to G418-SM agar plates (30 µg/ml G418) in the presence of G418 resistant *Ka* for a final round of selection. Stable clones were then mixed in equal proportions with their ancestor and allowed to develop. Those that did not significantly differ in proportion when mixed with their ancestor and allowed to develop were used in the assay (see Figs. S1 and S2).

## Cell preparation and migration assay

We washed harvested log-phase cells free of bacteria by repeated centrifugation and suspended them in KK2 buffer at a density of $1 \times 10^8$ cells/ml. We made 50:50 chimeric mixes of each RFP clone against all other ancestor clones, with a total of 10 chimeric mixes.

We placed 1.5% water agar Petri plates (size: 150 × 15 mm) in a laminar flow hood to remove excess moisture. We then drew a line on the underside of the plates that was 2 cm from the edge of the plate so that a line of $1 \times 10^7$ cells could be applied with accuracy and dried beneath a laminar flow hood for an additional 45 min. For each treatment there were 20 plates: ten chimeric mixes and ten clonal mixes (all 5 ancestors and their RFP-transformants).

We set up two different treatments: non-migration and migration. For the non-migration treatment, plates of each clone or mix were wrapped individually in foil with a 0.5 cm wide slit cut over the cells and then placed in an incubator where they could receive light from above, a condition which causes them to fruit without first migrating. For the migration treatment, plates were aligned and stacked with paper circles between each one. The plates were then wrapped in aluminum foil, leaving a small opening at the end of the plates opposite to the cells. This provided a directional light gradient for the aggregates to phototactically move toward. The plates from both treatments were incubated for 6 days in 24-hour light, before being unwrapped and placed beneath a unidirectional light source to induce fruiting of any slugs that remained. Each pair of treatments was replicated five times.

To measure migration distance, we followed the procedure in *Jack et al. (2011)* where the plate was marked in 2 cm wide zones parallel to the original line they were applied and counted the number of fruiting bodies per zone using a dissecting microscope.

## Estimation of spore allocation and rate of spore loss

### Spore production

Spore allocation was measured using spore production as a proxy. The fruiting bodies were carefully scraped up with a modified spatula and added to 3 mL of spore buffer (20 mM EDTA and 0.1% NP-40). To calculate spore production, the total number of spores was estimated and divided by the original number of cells.

To estimate the proportions of spores of both strains in a chimeric fruiting body, we counted the proportion of RFP-labeled cells using a fluorescent microscope, correcting

for loss of labeling from the clonally plated RFP genotypes. To reduce sampling error, we counted at least 250 spores.

### Rate of spore loss

We calculated the rate of spore lost as the decrease in spore production per centimeter traveled. We took the difference in spore production between the No Migration and Migration treatments and divided by the difference in distance traveled between the Migration and No Migration treatments. Standardizing for distance traveled allows us to accurately compare the proportion of spores that were lost for both treatments.

## Measuring cheating and facultative behavior

We calculated the spore production for each pair both clonally and chimerically following the procedure in (*Buttery et al., 2009*). Clonal spore production varies between genotypes. This is equivalent to fixed allocation cheating and must be accounted for when measuring facultative behavior. Facultative behavior is measured as the deviation from clonal spore production when in chimera. The amount of facultative behavior was calculated as the sum of the degree to which a genotype's own spore production increased ('self-promotion') and the amount it could reduce its competitors' ('coercion') during social competition. The values for coercion and self-promotion can be plotted as coordinates on a grid (Fig. 4). The origin stands for no change in behavior. Any deviation from the origin is considered facultative behavior. We compared the lengths of the vectors for the migration and no migration treatments.

## Morphometrics

We measured spore-stalk ratio directly from fruiting body architecture by estimating volumes. We calculated stalk volume using the average width of the stalk measured across the bottom, middle, and top of the stalk and the stalk length. We calculated spore allocation as the volume of the sorus divided by the volume of the whole fruiting body (*Buttery et al., 2009*). Seven or eight fruiting bodies from each clone or chimeric pair were measured.

## Statistical analysis

All statistical analyses were calculated using R software version 3.0 (www.r-project.org). Because of the 'round robin' nature of the experimental design, data were analyzed as nested ANOVAs, using 1-way or 2-way ANOVAs depending upon the number of factors in the analysis. This allowed us to control for variation between replicates.

## RESULTS

### Chimerism and migration

On average, slugs in the migration treatment traveled $5.76 \pm 0.017$ cm while slugs from the no migration treatment traveled $0.093 \pm 0.002$ cm (1-way nested ANOVA: $F_{6,214} = 476.02$, $P < 0.001$). We found that clonal genotypes vary in the distance they migrate (Fig. 1; 1-way nested ANOVA: $F_{4,34} = 7.55$; $P < 0.001$). Chimeric slugs migrated less far than would have been expected from the average of the migration distance of the two constituent

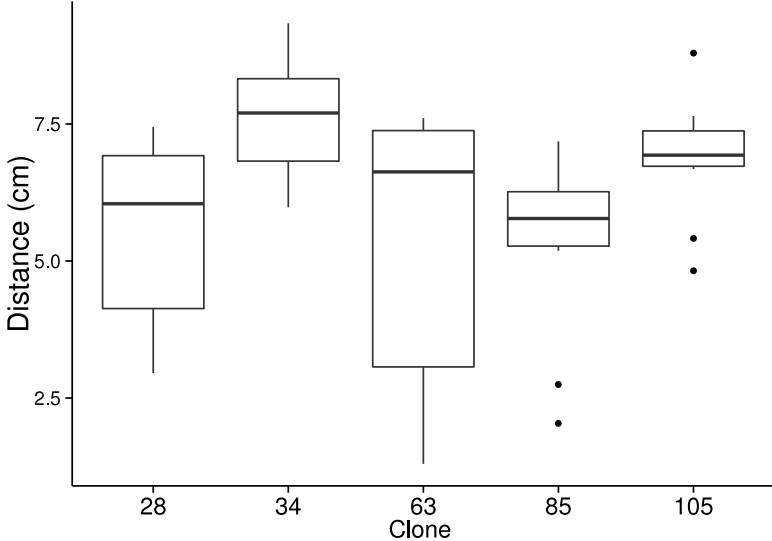

**Figure 1 Migration distance is genotype specific.** An equal number of cells of each clone was placed on water agar plates to form slugs. The slugs migrated under a unidirectional light source for six days and were then allowed to fruit. An average migration distance per plate was calculated. The Tukey boxplots shows the distribution of ten replicates (five untransformed, five RFP transformed) for each clone. (1-way nested ANOVA: $F_{4,34} = 7.55$, $P < 0.001$).

clones (Fig. 2A; observed mean = $5.50 \pm 0.24$ cm, expected mean = $6.19 \pm 0.20$ cm; 1-way nested ANOVA: $F_{1,49} = 17.86$, $P < 0.001$). When we calculated the decrease in spore production per centimeter traveled, we did not find a significant difference between clonal slugs ($\mu = -0.040 \pm 0.004$ spores per cell/cm) and chimeric slugs ($\mu = -0.046 \pm 0.005$ spores per cell/cm; Fig. 2B—one-way ANOVA; $F_{1,93} = 0.83$, $p = 0.365$).

## Migration affects spore production and allocation

We found that chimeric fruiting bodies contain more spores than clonal fruiting bodies, though the difference was only marginally significant. This confirms a previous significant result (*Buttery et al., 2009*). Chimeric fruiting bodies produced more spores compared to clonal fruiting bodies both with and without migration. The fruiting bodies of aggregates that migrated produced significantly fewer spores than those that did not migrate (Fig. 3A; 2-way nested ANOVA: clonal vs. chimeric: $F_{1,73} = 2.76$, $P = 0.066$, $\mu_{CL} = 0.214 \pm 0.011$ spores per cell, $\mu_{CH} = 0.247 \pm 0.017$ spores per cell; migration vs. non-migration: $F_{1,73} = 133.9$, $P < 0.001$, $\mu_M = 0.117 \pm 0.006$ spores per cell, $\mu_{NM} = 0.345 \pm 0.015$ spores per cell).

We found significant differences in fruiting body architecture between aggregates that migrated and those that did not. From the morphometric analysis of fruiting body structure, we found that fruiting bodies that migrated allocated proportionately more to spores than those that did not (Fig. 3B; 1-way nested ANOVA: $F_{1,24} = 10.46$, $P = 0.004$, $\mu_M = 0.933 \pm 0.01$, $\mu_{NM} = 0.836 \pm 0.014$). This was true for both clonal and chimeric aggregates. As expected from this result, aggregates that migrated had shorter stalks

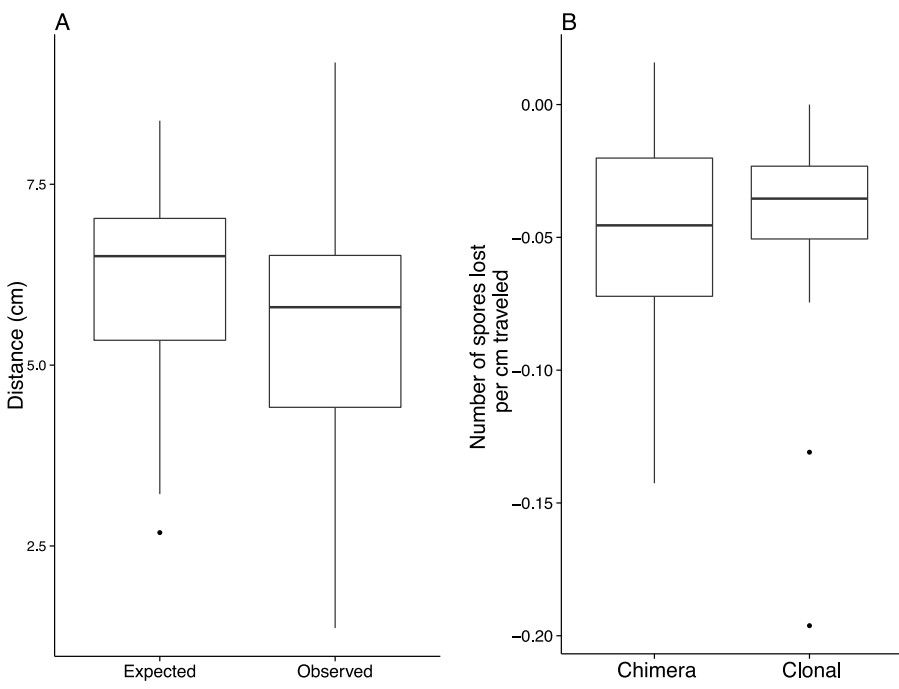

**Figure 2 Chimeric slugs travel less far than clonal slugs but lose cells over distance at a similar rate.** (A) Using the clonal migration distances from Fig. 1, we calculated the expected migration distances for chimeric slugs that developed from the same total number of cells. The Tukey boxplots show that migration distance for chimeric slugs were lower compared to clonal slugs (observed mean = 5.50 ± 0.24 cm, expected mean = 6.19 ± 0.20; 1-way nested ANOVA: $F_{1,49} = 17.89$, $P < 0.001$). (B) However, the decreased migration did not seem to affect spore production as there was not a significant difference in the number of spores lost per cm traveled between clonal and chimeric fruiting bodies after migrating (1-way nested ANOVA: $F_{1,93} = 0.83$, $P = 0.365$).

than those that did not (Fig. 3C; 1-way nested ANOVA: $F_{1,60} = 804.1$, $P < 0.0001$, $\mu_M = 311.24 \pm 11.71$ mm, $\mu_{NM} = 1046.47 \pm 27.23$ mm).

### Migration causes a decrease in cheating behavior

We estimated the amount of fixed cheating and facultative cheating between the two treatments by comparing the spore production of clonal and chimeric fruiting bodies. We found no differences in relative fixed allocations (i.e., clonal spore allocation) when we compared spore allocation with and without migration. However, there was significantly less facultative cheating behavior within chimeras that migrated compared to those that did not (Fig. 4; 1-way ANOVA: $F_{1,22} = 22.18$, $P < 0.001$, $\mu_M = 0.086 \pm 0.014$, $\mu_{NM} = 0.175 \pm 0.02$).

## DISCUSSION

Previous studies have found cheating in *D. discoideum* and that it can be divided into two categories: fixed and facultative (*Strassmann, Zhu & Queller, 2000*; *Fortunato, Queller & Strassmann, 2003*; *Queller et al., 2003*; *Gilbert et al., 2007*; *Buttery et al., 2009*). The proportion of cells allocated to spore vs. stalk is generally a genotype-specific trait, so if a high spore allocator is mixed with a low spore allocator, the high spore allocator is

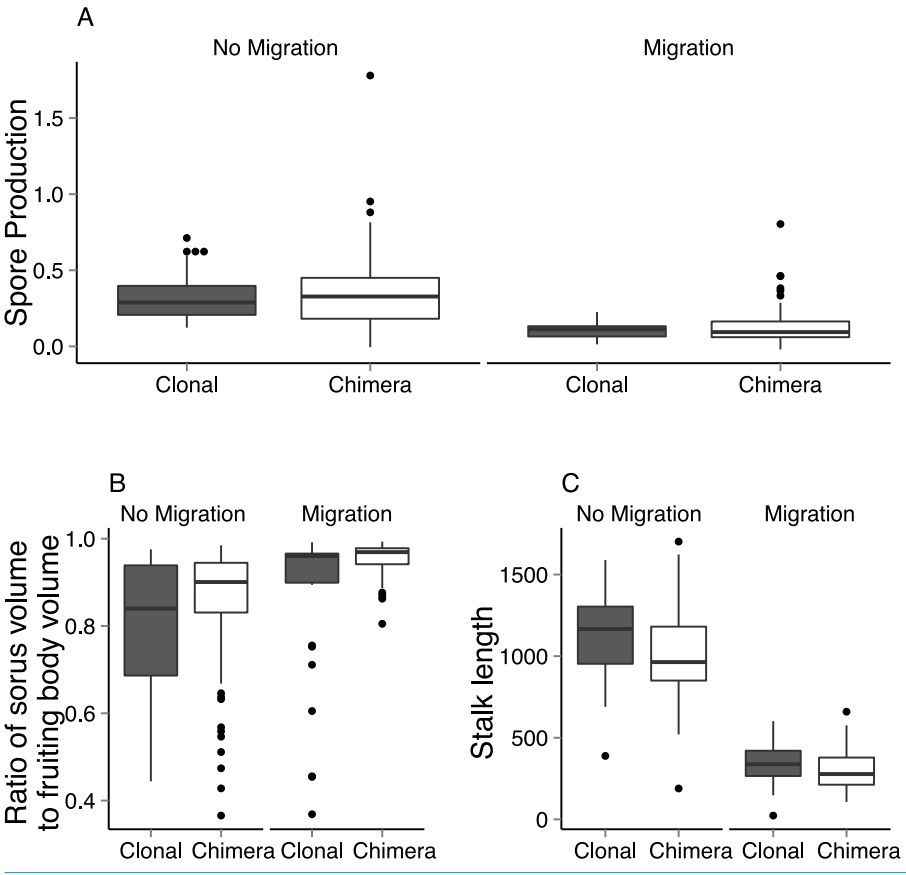

**Figure 3 Spore production and fruiting body architecture is affected by migration and whether fruiting bodies are clonal or chimeric.** The Tukey boxplots compare different measurements of fruiting body production between groups and treatments. (A) This shows that clones that migrated had a significantly lower spore production than fruiting bodies that did not, indicating the loss of cells as the slugs migrated. Chimeric fruiting bodies had a higher, marginally significant, spore production compared to clonal fruiting bodies across both *non-migration* and *migration* treatments (2-way nested ANOVA: non-migration vs. migration: $F_{1,73} = 133.9$, $P < 0.001$; clonal vs. chimeric: $F_{1,73} = 2.76$, $P = 0.063$). (B) The ratio of sorus volume to total fruiting body volume of migrated fruiting bodies are significantly higher compared to those of the non-migration treatment, irrespective of whether the fruiting bodies were clonal or chimeric (1-way nested ANOVA: $F_{1,24} = 10.46$, $P = 0.004$). (C) The higher ratio of sorus to fruiting body shown in B may be explained because fruiting bodies that have migrated have significantly shorter stalks than those that did not migrate (1-way nested ANOVA: $F_{1,60} = 804.1$, $P < 0.0001$).

expected to be overrepresented in the sorus. This is fixed cheating, and the degree to which it occurs can be predicted from genotypes' clonal behavior (*Buttery et al., 2009*). Facultative cheating occurs when there is a significant deviation from the behavior exhibited under clonal conditions. Genotypes that cheat by increasing their own allocation to spores are 'self-promoters' and those that can reduce their partner's share are 'coercers' (*Buttery et al., 2009*; *Parkinson et al., 2011*). Partitioning cheating behaviors have given us a lot of new insight in kin conflict in *D. discoideum*, but these studies are limited because they do not focus on competition during the migration stage, which makes up a large portion of the social life cycle.

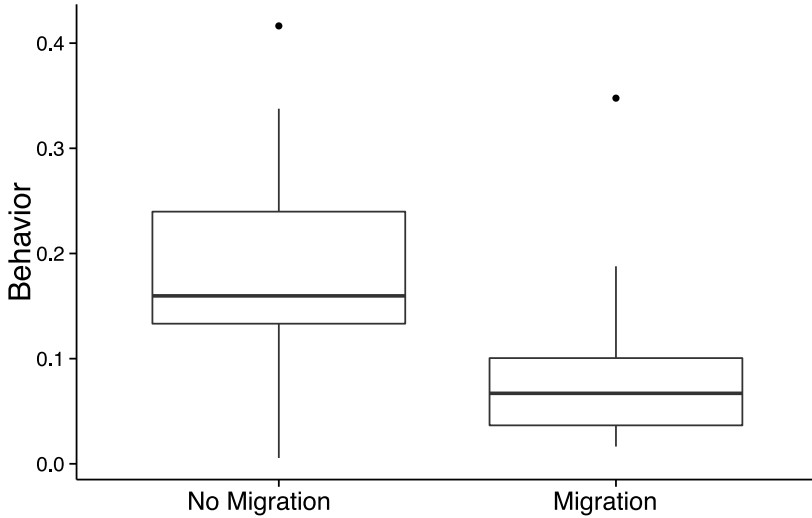

**Figure 4 Facultative cheating behavior is reduced after migration.** Facultative cheating, the deviation from clonal spore production when in chimera is the sum of 'self-promotion' and 'coercion,' is shown in the Tukey boxplots as their overall behavior. Overall, this cheating behavior decreased by approximately 50% for fruiting bodies that migrated compared to those that did not (1-way nested ANOVA: $F_{1,22} = 22.18$, $P < 0.001$).

In this study, we examined the interplay of two seemingly diametric actions, the solitary action of kin recognition and the collective action of slug migration in *D. discoideum*, to more fully understand the effects of social competition on fitness over the entire lifecycle. The study by *Foster et al. (2002)* found that chimeric slugs did not travel as far as clonal slugs of the same size. They hypothesized that internal conflict was preventing the slugs from traveling greater distances. The anatomy of the *Dictyostelium* slug is such that the front of the slug is where the cells that will eventually become stalk are located. They suggested that the unwillingness to be in the front of the slug might be the cause of the shorter distances. More recent studies suggest that response to DIF-1, a polyketide produced by prespore cells that induce differentiation into stalk, can predict whether a clone is likely to cheat or be a cheater (*Parkinson et al., 2011*). Clones that were more sensitive to DIF-1 were more likely to end up in the stalk. Our initial hypothesis was that if social competition is prolonged by migration towards light, the behavior of cheaters would be exaggerated if cheating is an active process where clones can either change their behavior if they sense a competitor or change the behavior of their competitor. We predicted that the lower relatedness of the chimeric slugs would increase the conflict within the slug, thus decreasing the probability of cells working as a cohesive unit to migrate and increasing the fitness of cheater clones.

Overall, our hypothesis was not supported. Although we did find a cost in distance traveled when we compared chimeric and clonal slugs, the difference was not nearly as large as that as in the study by *Foster et al. (2002)*. *Castillo et al. (2005)* showed that slugs found within shallow soil (1 cm from the surface) could easily travel to the surface, whether or not they were chimeric and that neither clonal nor chimeric slugs

could easily reach the surface when under a 5 cm-deep layer of soil. Additionally, cells sloughed during migration will have seeded new colonies should the slug pass through a patch of bacterial food (*Kuzdzal-Fick et al., 2007*), so even if the slug is unable to make it completely through to the soil surface, the cells from the slug will still have the opportunity to replicate. Most interestingly, instead of finding increased cheating, the outcome of our interactions showed a decrease in cheating behavior when chimeric slugs were allowed to migrate compared to when they were not. Weaker clones from the no migration treatment had increased spore representation in the migration treatment suggesting that migration reduced the costs associated with being a chimeric slug. There are two possible explanations for our results. A recent paper found that kin recognition is lost during the slug stage and that kin discrimination and cheating both decrease as development proceeds (*Ho & Shaulsky, 2015*). Slug migration lengthens the development time, as *D. discoideum* does not begin differentiating until it has reached a new location. It is possible that the decrease in facultative cheating is related to the tgrB1 and tgrC1 genes decreased expression levels, which leads to less kin recognition. Another possible explanation is related to the production of DIF-1. Those clones that migrated the farthest (Fig. 1) were also the clones that were most likely to be facultative cheaters according to (*Parkinson et al., 2011*). These clones are the ones that show the least response to DIF-1 and produced the most. If in chimeras, these longer migrating clones are no longer at the front, it could explain why chimeric slugs travel shorter distances than clonal slugs. Additionally, it is possible that the act of migration is energetically costly, so that these clones produce less DIF-1. If that were the case, then clones that are more sensitive to it under non-migration circumstances would show increased spore production, which would give the results that we saw—more equitable distribution of spores.

When we compared our morphometric analysis of fruiting bodies for all treatments, we found another consequence of migration. We found that spore-stalk allocation increases with migration for both clonal and chimeric treatments. This may be a non-adaptive response to the decreased DIF-1 production. Or, producing proportionally less stalk after prolonged migration may be a useful strategy; stalk height may be less important if the slug has migrated into a more suitable habitat for dispersal. Dictyostelid spores are sticky, and therefore not likely to be dispersed by wind, but viable spores from dictyostelids have been found in the digestive contents of earthworms, nematodes, and other soil invertebrates, which can act as mid-distance dispersers or can travel over even longer distances in the digestive tracts of birds and mammals (*Suthers, 1985*; *Huss, 1989*; *Sathe et al., 2010*).

## CONCLUSIONS

Collective cell and animal behavior is useful for understanding the evolution of multicellularity. Migration in *D. discoideum* encompasses concepts from both types of behavior. Collective cell migration is necessary for two of the key processes of embryonic development: gastrulation and organogenesis (*Weijer, 2009*). Cell migration in *Dictyostelium* is very similar (*Weijer, 2009*). Both involve cells that are close together, migrate easily, move collectively in response to a signal, use actin and cell–cell junctions to

provide traction, and have an extracellular matrix (*Friedl & Gilmour, 2009*; *Weijer, 2009*). Collective animal behaviors such as grouping and swarming involve self-organization and are found in both lower and higher organisms (*Sumpter, 2006*; *Olson et al., 2013*). They provide many benefits such as reducing the risk of predation, increase foraging efficiency, and improving mating success (*Olson et al., 2013*). For *Dictyostelium*, collective migration allows the cells to move more efficiently and for longer distances than individuals, much like the V formation in migrating geese. However, there are instances where individuals within a group may go rogue and only think of their own self-interest, such as when cheaters gain more of a public good than they contribute. Slug migration is beneficial to all cells because it aligns the interests of the cells towards migration. Our study suggests that migration may also lead to alleviation of the conflict of interests in heterogeneous slugs, which leads to a decrease in facultative cheating.

## ACKNOWLEDGEMENTS

We thank members of the Strassmann-Queller lab for many helpful discussions about experimental design, setup, and data analysis.

### Funding

This material is based upon work supported by the National Science Foundation under grant numbers IOS1256416 and DEB1146375 and the John Templeton Foundation. The funders had no role in study design, data collection and analysis, decision to publish, or preparation of the manuscript.

### Grant Disclosures

The following grant information was disclosed by the authors:
National Science Foundation: IOS1256416, DEB1146375.
John Templeton Foundation.

### Competing Interests

The authors declare there are no competing interests.

### Author Contributions

- Chandra N. Jack conceived and designed the experiments, performed the experiments, analyzed the data, wrote the paper, prepared figures and/or tables, reviewed drafts of the paper.
- Neil Buttery conceived and designed the experiments, performed the experiments, analyzed the data, wrote the paper, reviewed drafts of the paper.
- Boahemaa Adu-Oppong and Michael Powers performed the experiments.
- Christopher R.L. Thompson contributed reagents/materials/analysis tools, reviewed drafts of the paper.

- David C. Queller and Joan E. Strassmann conceived and designed the experiments, contributed reagents/materials/analysis tools, wrote the paper, reviewed drafts of the paper.

## Supplemental Information

Supplemental information for this article can be found online at http://dx.doi.org/10.7717/peerj.1352#supplemental-information.

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
