# Peer review of "Migration in the social stage of Dictyostelium discoideum amoebae impacts competition"

_PeerJ, doi:10.7717/peerj.1352_

## Round 0.1 · original submission · Major Revisions

In addition to addressing the reviewer comments, please consider revising your figures along the lines suggested in this recent PLOS Biology article: http://journals.plos.org/plosbiology/article?id=10.1371/journal.pbio.1002128

It seems to me that your data set would be suitable for and benefit from this visualization approach.

Reviewer 1 ·

Basic reporting

original data are not provided. (as far as I can see).
Should be on principle - but in this case they are even more useful and not that large
. I would prefer a table in main text.
Given the data consistency rather than significant average differences can be checked, as well as now unsupported statement as "weaker clones...(r 233-234)
as no clone specific outcomes are reported.

More quantitative assessment of the finding in the text (e.g. the overwhelming incfluence of migration) should be reported as such.

Experimental design

I am not qualified to judge the appropriateness of (pre)treatments of the clones, and wrap-ups of the slugs

Validity of the findings

I assume the findings are valid.
The exclusive reporting of the findings in terms of cooperation, competition and cheating, rather than hypothesizing in what mechanical way the results can be explained, e.g. how could migration and spore formation be causally linked?
(e.g. in terms of adhesion and position dependent redifferentiation) I find less compelling (see e.g. the account of one of the authors Queller 2003
that knock-out of adhesion molecule leads to more spores, and the hazards thereof in the accumulation phase.) Could differential adhesion, or differences in chemotaxis
lead to differential loss of cells. etc.

The ininital (disproved) hypothesis of longer competition in the migrating slug
compels the question competition for what?

Reviewer 2 ·

Basic reporting

In this very interesting manuscript, the authors are investigating the effect of migration (a collective behavior), on cooperation vs. defection in chimeric as opposed to clonal slugs of Dicty. Studying this interaction is important as the migration stage can have significant effects on possible sorting mechanisms that precede the actual competition. The authors find that migration reduces conflicts, and decreases the costs of being associated with a chimeric slug. (Note that the manuscript says “cheating costs”, but there are only cheating costs to the group, not the individual. I would have said that it either increases the benefits, or decreases the cost to the individual. There is no cost of cheating to the individual.)

Overall, I find the study well constructed, and the results are presented clearly. But the conclusions are perhaps less clear. Fig. 4 shows that migrating slugs have shorter stalk length, and this indeed points towards lower costs to the individual. I say perhaps, because the cost/benefit analysis really should be made taking into account fraction in sorus to fraction in stalk. We can read this off from Fig. 3B, and it does show a clear increase in benefit/cost (cost is being left in the stalk, benefit is going towards spore). But of course, all these costs and benefits are modulated by the ability to travel, and therefore cannot easily be calculated in a model.

This data alone does not support the notion that there is less conflict in the migrating slug (less cheating). But the data on facultative cheating does, but it throws no light on the differences between clonal and non-clonal slugs, as you cannot measure a change in facultative cheating between clonal and non-clonal treatment as this difference is made in reference to clonal slugs. In fact, most of the evidence points towards the conclusion that it does not matter whether the slug is clonal or not. For this to be the case, one would require a mechanism to exclude cheaters, meaning that their facultative self-promotion or coercion is punished, for example by not providing stalks to those. But this is highly risky, because such punishment must occur during the migration stage, so as not to also punish conspecifics. After all, they are all in one boat. There is some evidence that such sorting-out of cheaters during the migration stage happens, but none is presented here. The evidence is usual molecular in nature, such as a particular adhesion molecule present in cooperators, but a different variant in cheaters. Also, it appears that fruiting bodies are mostly clonal even if the slug is chimeric (or so I have read), which also points to sorting during migration. None of this is discussed here, but I think it is to some extent the crucial part in understanding the interplay between migration and altruism. If the authors could address this issue more forcefully, this would strengthen the paper tremendously.

And again, it makes no sense to say that you predicted that lower relatedness of chimeric slugs would increase the conflict in the slug, when conflict is only measured by facultative cheating, which cannot be measured in clonal slugs. Migration obviously reduces conflicts because it reduces the benefit to the cheaters. But this has nothing to do with relatedness. I would like to know how Dicty manages to exclude potential cheaters during migration. I’m not even sure that there is such evidence in this paper (but there certainly is in previous papers by the group). It is true that migration alleviates the conflicts of interest of competitors in a chimera by aligning their interest. But you still need to exclude cheaters who come to ride on the same boat. And I don’t see here how this is achieved.

Experimental design

No Comments

Validity of the findings

No Comments

Additional comments

I find this line of investigation compelling, and the system provides an ideal experimetnal platform to study the interplay between selfish decisions and collective behaviour. For me, it raised more questions than it answered, but perhaps that is because these are important and compelling questions to begin with.

---

## Round 0.2 · Minor Revisions

I agree with the reviewer that some additional clean-up and re-wording would improve the manuscript.

Reviewer 2 ·

Basic reporting

I am satisfied with the author's attempts at discussing possible molecular mechanisms that link reduced conflict, migration, and heterogeneity. However the manuscript still says "migration reduced the cheating costs associated with being a chimeric slug", which seems to imply that there is a cost of cheating to the individual. There is not. A simple rewording will probably do. I don't understand why this was never addressed by the authors.

I also appreciate the new figure format, but the figure captions really should say more. They are extremely terse, and do not explain the nature of the whiskers in the box plots: there are different styles, and a caption must identify what they are.

RFP: must give abbreviation. I know what it is, but isn’t there a rule that all acronyms must redefined in the text?

For all of the statistical tests the authors quote F-tests. That’s appropriate for ANOVA with more than two groups, but for two groups the test should be equivalent to a two simple non-equal variances t-test. I redid one of the tests (for Fig. 3B), assuming that the given error bars are 95% confidence intervals (this is not stated). I get t=3.19, which is consistent with F=t^2 for two groups. A reader might conceivably ask why a t-test is not used, when in fact they are the same thing.

I could also not find any mention in the text of the number of replicates for each treatment. I could find them in the Excel file, but that file has no description associated with it, so it is really a little bit of detective work to dig that out. It should not be too hard to amend the Methods to discuss replicates, and write a description file for the Excel sheets.

Experimental design

No comments

Validity of the findings

No comments

Additional comments

No comments

---

## Round 0.3 · accepted · Accept

Thank you for addressing the remaining comments.